# TURBO: The Swiss Knife of Auto-Encoders

**DOI:** 10.3390/e25101471

**Published:** 2023-10-21

**Authors:** Guillaume Quétant, Yury Belousov, Vitaliy Kinakh, Slava Voloshynovskiy

**Affiliations:** Centre Universitaire d’Informatique, Université de Genève, Route de Drize 7, CH-1227 Carouge, Switzerland; yury.belousov@unige.ch (Y.B.); vitaliy.kinakh@unige.ch (V.K.)

**Keywords:** information bottleneck, TURBO, generalisation, auto-encoder, variational approximation, lower bound, mutual information, physical latent space, representations, Kullback–Leibler divergence

## Abstract

We present a novel information-theoretic framework, termed as TURBO, designed to systematically analyse and generalise auto-encoding methods. We start by examining the principles of information bottleneck and bottleneck-based networks in the auto-encoding setting and identifying their inherent limitations, which become more prominent for data with multiple relevant, physics-related representations. The TURBO framework is then introduced, providing a comprehensive derivation of its core concept consisting of the maximisation of mutual information between various data representations expressed in two directions reflecting the information flows. We illustrate that numerous prevalent neural network models are encompassed within this framework. The paper underscores the insufficiency of the information bottleneck concept in elucidating all such models, thereby establishing TURBO as a preferable theoretical reference. The introduction of TURBO contributes to a richer understanding of data representation and the structure of neural network models, enabling more efficient and versatile applications.

## 1. Introduction

Over the past few years, many deep learning architectures have been proposed and successfully applied to a wide range of problems. However, they are often developed from empirical observations and their theoretical foundations are still not well enough understood. Typical examples of deep learning architectures that have been widely used and revisited multiple times are generative adversarial networks (GANs) [1], variational auto-encoders (VAEs) [2,3] and adversarial auto-encoders (AAEs) [4]. A rigorous interpretation of the concepts and principles behind such machine learning methods is crucial to understanding their strength and limitations, and to guiding the development of new models. A concrete formulation of these concepts, unifying as many models as possible, would be a huge gain for the field.

Showing a promising path towards this goal, bottleneck formulations of neural network training have been extensively studied in many theoretical and experimental works [5,6,7,8,9]. They are based on the fact that one may want to preserve as much relevant information as possible from a given input, removing all unnecessary knowledge. This is called the information bottleneck (IBN) principle [10] and it has a crucial impact in several applications.

It was originally developed to characterise what can be termed “relevant” information in the context of supervised learning [10]. The IBN principle was introduced as an extension to the so-called *rate-distortion* theory [11], leaving the choice of the distortion function open and giving an iterative algorithm for finding the optimal compressed representation of the data. This first description of the IBN principle paves the way for machine-learning-oriented formulations of supervised learning [5], as well as for the bounded information bottleneck auto-encoder (BIB-AE) framework [12] in the context of unsupervised learning. The BIB-AE framework shows that many bottleneck architectures, especially concerning the VAE family, are generalised by this approach. It can also be used for semi-supervised learning [7], where the framework allows the impact of several well-known techniques to be studied in a better and more interpretable way. More recently, the IBN principle has been used as an attempt to explain the success of self-supervised learning (SSL) [13,14]. In this context, the bottleneck manifests itself as a compression of information between the learnt representation and the distorted image, usually obtained by applying augmentations to the original input. This is achieved by minimising the mutual information between the representation and the distorted image, while maximising the mutual information between the representation and the original image. However, a key limitation rises from this formulation of SSL, since it is largely based on the assumption that all the relevant information for a given downstream task is shared by both the original and the distorted images. If this is not the case, which may occur whenever one replaces the distorted image by a second meaningful representation of the data, the IBN principle struggles to provide a satisfactory explanation of SSL.

The IBN principle represents a significant theoretical advance towards explainable machine learning, but it has several inherent limitations [8]. For example, while the IBN provides a solid and comprehensive theoretical interpretation for the VAE family of models, it can only address the GAN family with an intricate formulation and it encounters difficulties in providing a compelling justification for the AAE family. The core issue emerges when the objective is not to achieve maximum disentanglement of the latent space from the input data, a scenario that is particularly salient with AAE models. The training of an AAE encoder is oriented to maximise the informational content shared between the input data and the latent representation, in direct contradiction to the premises of the IBN principle. This shift in approach is not simply a mathematical manoeuvre to articulate new models: it fundamentally alters the desired methodology of data interpretation. When dealing with input data that admit two or more relevant representations, and where one representation is designated as a physically meaningful latent space, it becomes natural to construct a framework that maximises the mutual information between these two highly correlated modalities.

On top of AAE type models, several other architectures cannot be interpreted via the IBN principle, but rather need a new paradigm. Image-to-image translation models such as pix2pix [15] and CycleGAN [16] played an important role in the development of machine learning. As we will demonstrate, they directly fall into the category of models that maximise the mutual information between two representations of the data, which is a case that the IBN principle fails to address. Furthermore, normalising flows [17] suffer from the same intricate formulation as GANs when attempting to explain them in terms of the IBN principle. Additionally, other models such as ALAE [18] do not show a bottleneck architecture and are thus unconvincingly interpretable via the IBN. All these points speak for the necessity of developing a new framework capable of addressing and explaining these methods theoretically.

In this work, we present a powerful formalism called Two-way Uni-directional Representations by Bounded Optimisation (TURBO) that complements the IBN principle, giving a rigorous interpretation of an additional wide range of neural network architectures. It is based on the maximisation of the mutual information between various random variables involved in an auto-encoder architecture.

The structure of the paper is following this logic: in Section 2, we define a unified language in order to make the understanding and the comparison of the various frameworks clearer. In Section 3, we explain the BIB-AE framework using these notations, exposing its main advantages and drawbacks, and the reasons for considering the TURBO alternative. In Section 4, we detail the TURBO framework and its generalisation power. In Section 5, we highlight successful applications of TURBO for solving several practical tasks. For the ease of reading, Table 1 shows a summary comparison of the BIB-AE and the TURBO frameworks. The reader who is already familiar with the IBN principle is advised to proceed directly to Section 3.2.

Our main contributions in this work are:Highlighting the main limitations of the IBN principle and the need for a new framework;Introducing and explaining the details of the TURBO framework, and motivating several use cases;Reviewing well-known models with the lens of the TURBO framework, showing how it is a straightforward generalisation of them;Linking the TURBO framework to additional related models, opening the door to additional studies and applications;Showcasing several applications where the TURBO framework gives either state-of-the-art or competing results compared to existing methods.

## 2. Notations and Definitions

Before discussing the various frameworks that we study in this work, we define a common basis for the notations. Since most of the considered models will be viewed as auto-encoders or as parts of auto-encoders, we shape our notations to fit this framework. Most of the quantities found throughout the paper are defined below and a table summarising all symbols and naming used can be found in Appendix A.

We consider two random variables X and Z with marginal distributions X∼p(x) and Z∼p(z), respectively, and either a known or unknown joint distribution p(x,z)=p(x|z)p(z)=p(z|x)p(x). Notice that X and Z can be independent variables, in which case their joint distribution simplifies to the product of their marginal distributions. The two unknown conditional distributions can be parametrised by two neural networks, usually called *encoder* $q_{\phi }\left(z|x\right)\approx p\left(z|x\right)$ and *decoder*
$p_{\theta }\left(x|z\right)\approx p\left(x|z\right)$, where the parameters of the networks are generically denoted by ϕ and θ. Once chained together as shown in Figure 1, they form a so-called *auto-encoder*. We thus define two approximations of the joint distribution
(1)$$\begin{array}{cc} q_{\phi }\left(x,z\right) & :=q_{\phi }\left(z|x\right)\;\overset{\overset{\mathrm{real}}{\mathrm{data}}}{\overset{︷}{p(x)}}=q_{\phi }\left(x|z\right)\overset{\overset{\mathrm{synthetic}}{\mathrm{data}}}{\overset{︷}{{\tilde{q}}_{\phi }\left(z\right)}}, \end{array}$$
(2)$$\begin{array}{cc} p_{\theta }\left(x,z\right) & :=\underset{\underset{\mathrm{networks}}{\mathrm{known}}}{\underset{︸}{p_{\theta }\left(x|z\right)}}\;p\left(z\right)=\underset{\underset{\mathrm{networks}}{\mathrm{unknown}}}{\underset{︸}{p_{\theta }\left(z|x\right)}}\;\;{\tilde{p}}_{\theta }\left(x\right)\;\;, \end{array}$$
where the rightmost expressions are reparametrisations with unknown networks and the approximated marginal distributions ${\tilde{q}}_{\phi }\left(z\right)=\int q_{\phi }\left(x,z\right)dx$ and ${\tilde{p}}_{\theta }\left(x\right)=\int p_{\theta }\left(x,z\right)dz$ corresponding to the latent spaces synthetic variables. Two approximated marginal distributions, also used throughout this work, corresponding to the reconstructed spaces synthetic variables can be defined as ${\hat{q}}_{\phi }\left(z\right)=\int {\tilde{p}}_{\theta }\left(x\right)\;q_{\phi }\left(z|x\right)dx$ and ${\hat{p}}_{\theta }\left(x\right)=\int {\tilde{q}}_{\phi }\left(z\right)\;p_{\theta }\left(x|z\right)dz$.

Since mutual information between different variables is extensively used in this paper, we give a brief definition of it. We also showcase in Figure 2 the notations that we employ when computing the mutual information between diverse random variables, corresponding to diverse information flows in the networks. The mutual information for two random variables X and Z following a joint distribution p(x,z) is defined as
(3)$$I\left(X;Z\right):=E_{p(x,z)}\left[\log\frac{p(x,z)}{p(x)p(z)}\right],$$
where E[·] is the mathematical expectation with respect to the given distribution p(x,z) and where p(x) and p(z) denote the corresponding marginal distributions. Therefore, to exemplify our notations, the mutual information between X and the random variable $\tilde{Z}$ defined by the marginalisation of the encoder $q_{\phi }\left(z|x\right)$ outputs $\tilde{Z}∼{\tilde{q}}_{\phi }\left(z\right)$ would be defined by $I_{\phi }\left(X;\tilde{Z}\right)=E_{q_{\phi }\left(x,z\right)}\left[\log q_{\phi }\left(x,z\right)/p\left(x\right){\tilde{q}}_{\phi }\left(z\right)\right]$. On the other hand, in order to compute the mutual information between the random variable defined by the marginalisation of the decoder $p_{\theta }\left(x|z\right)$ outputs $\tilde{X}∼{\tilde{p}}_{\theta }\left(x\right)$ and Z, its expression would be $I_{\theta }\left(\tilde{X};Z\right)=E_{p_{\theta }\left(x,z\right)}\left[\log p_{\theta }\left(x,z\right)/{\tilde{p}}_{\theta }\left(x\right)p\left(z\right)\right]$.

Lastly, we use several other common information-theoretic quantities such as the Kullback–Leibler divergence (KLD) between two distributions denoted by $D_{\mathrm{KL}}\left(\cdot ∥\cdot \right)$, the entropy of a distribution denoted by H(·), the conditional entropy of a distribution denoted by H(·|·) and the cross-entropy between two distributions denoted by H(·;·).

## 3. Background: From IBN to TURBO

In this section, for the completeness of our analysis, we briefly review the BIB-AE framework [12], which addresses a variational formulation of the IBN principle for an auto-encoding setting. We then redefine the two founding models at the root of many deep learning studies, the so-called VAE and GAN, in order to unite them under the BIB-AE umbrella and our notations. Finally, we explain why this formulation, even though well suited to many problems and showing several advantages, is not applicable to applications with a physical latent space where the data compression is not needed as such.

### 3.1. Min-Max Game: Or Bottleneck Training

The IBN is based on a compression principle where all targeted task-irrelevant information should be filtered out from the input data, i.e., the input data are *compressed*. The difference with the classical compression addressed in the rate-distortion theory, which ensures the best source reconstruction under a given rate, is that the IBN compressed data contain only sufficient statistics for a given downstream task. The main targeted application of the IBN is classification [5,19], where the intermediate or latent representation contains only the information related to the provided class labels. Recently, the IBN has also been extended to cover a broad family of privacy-related issues through the complexity–leakage–utility bottleneck (CLUB) [20]. For example, it can be used to disentangle some sensitive data from a latent representation in such a way that no private information remains, while useful features are still available for downstream tasks. Bottleneck networks are also used in anomaly detection in order to get rid of all the useless features contained in the data, keeping only what helps in identifying background from signal [21,22,23]. A parallel usage of such models is for the generation of new data from a given manifold [24,25]. Indeed, as the encoded latent space should ideally be a disentangled representation of the data, it might be shaped towards any desired random distribution. Typically, this distribution would be Gaussian so that one can generate new samples by passing Gaussian noise through the trained decoder, which should have learnt to recover the data manifold from the minimal amount of information kept by such latent representation. More generally, formalisms derived from the IBN principle describe ways to create a mapping between some input data and a defined distribution.

The BIB-AE framework formulated for both unsupervised [12] and semi-supervised settings [7] is a variational formulation of the IBN principle [5]. It relies on a trade-off between the amount of information lost when encoding data into a latent space and the amount of information kept for proper decoding from this latent space. It can be formally expressed as an optimisation problem where one tries to balance minimisation and maximisation of the mutual information between the data space random variable X and the latent space random variable $\tilde{Z}$. Therefore, the BIB-AE loss has the form
(4)$$L^{\mathrm{BIB}-\mathrm{AE}}\left(\phi ,\theta \right)=I_{\phi }\left(X;\tilde{Z}\right)-\lambda_{B}\;I_{\phi ,\theta }^{x}\left(X;\tilde{Z}\right),$$
where $I_{\phi ,\theta }^{x}\left(X;\tilde{Z}\right)$ is a parametrised lower bound to $I_{\phi }\left(X;\tilde{Z}\right)$ as demonstrated in Appendix B. The positive weight $\lambda_{B}$ controls the trade-off between the minimisation and maximisation of the two terms. Once Equation (4) expands, the resulting loss becomes
(5)$$\begin{array}{cc} L^{\mathrm{BIB}-\mathrm{AE}}\left(\phi ,\theta \right)= & E_{p(x)}\left[D_{\mathrm{KL}}\left(q_{\phi }\left(z|x\right)∥p\left(z\right)\right)\right]-D_{\mathrm{KL}}\left({\tilde{q}}_{\phi }\left(z\right)∥p\left(z\right)\right)\\ \; & -\lambda_{B}\;E_{q_{\phi }\left(x,z\right)}\left[\log p_{\theta }\left(x|z\right)\right]+\lambda_{B}\;D_{\mathrm{KL}}\left(p\left(x\right)∥{\hat{p}}_{\theta }\left(x\right)\right). \end{array}$$Notice that we intentionally abuse the notation of $D_{\mathrm{KL}}\left(q_{\phi }\left(z|X=x\right)∥p\left(z\right)\right)$ here and in other analogous expressions in order to lighten the equations since the expectations always solve the ambiguity. The full derivation of Equation (5) is given in Appendix B. Next, we showcase how the BIB-AE formalism can be related to the VAE and GAN families.

#### 3.1.1. VAE from BIB-AE Perspectives

In the wide range of machine learning methods, VAE [2,3] has a particular place as it is among the first deep learning generative models developed. VAE has been extensively studied to decorrelate a given data space, mapping it to a Gaussian latent distribution while reconstructing back the data space from it. This is a typical example of the IBN principle and therefore it easily falls under the BIB-AE formalism. Keeping only the first and third terms of Equation (5) directly leads to the VAE loss
(6)$$L^{\mathrm{VAE}}\left(\phi ,\theta \right)=E_{p(x)}\left[D_{\mathrm{KL}}\left(q_{\phi }\left(z|x\right)∥p\left(z\right)\right)\right]-\lambda_{B}\;E_{q_{\phi }\left(x,z\right)}\left[\log p_{\theta }\left(x|z\right)\right],$$
where we already allow for the weight $\lambda_{B}$ to appear in order to generalise to a broader family called β-VAE [26]. In the literature, the $\lambda_{B}$ weight is often written β, hence the name β-VAE. The usual VAE loss would correspond to $\lambda_{B}=1$.

Typically, the encoder is designed in such a way as to output two values that are used to parametrise the mean and the variance of a Gaussian distribution from which latent points are drawn. Therefore, the conditional latent space distribution $q_{\phi }\left(z|x\right)$ is a conditional Gaussian distribution. For p(z) chosen to be a standard normal distribution, the KLD term has a closed form and can be exactly and efficiently computed. For the second term of the loss, it is typical to assume that the conditional decoded space shows exponential deviations from the original data leading to $\log p_{\theta }\left(x|z\right)=-\alpha {∥\hat{x}-x∥}_{p}^{p}+C$, where α and *C* are positive constants.

A successful extension called information maximising VAE (InfoVAE) [27] proposes to add the mutual information term $I_{\phi }\left(X;\tilde{Z}\right)$ to the VAE objective and to maximise it. This is exactly the first term of the BIB-AE loss in Equation (4), which is, however, minimised, so that the InfoVAE objective is actually counter-balancing it. The expression of the InfoVAE loss is given by the first, second and third terms of Equation (5) as well as an additional weight factor $\lambda_{\mathrm{Info}}$
(7)$$\begin{array}{cc} L^{\mathrm{InfoVAE}}\left(\phi ,\theta \right)= & E_{p(x)}\left[D_{\mathrm{KL}}\left(q_{\phi }\left(z|x\right)∥p\left(z\right)\right)\right]-\left(1-\lambda_{B}\lambda_{\mathrm{Info}}\right)D_{\mathrm{KL}}\left({\tilde{q}}_{\phi }\left(z\right)∥p\left(z\right)\right)\\ \; & -\lambda_{B}\;E_{q_{\phi }\left(x,z\right)}\left[\log p_{\theta }\left(x|z\right)\right], \end{array}$$
where the usual VAE loss would correspond to $\lambda_{B}=\lambda_{\mathrm{Info}}=1$. Notice that, in the original InfoVAE loss [27], the weights are denoted by α and λ, with $\lambda_{B}=1/\left(1-\alpha \right)$ and $\lambda_{\mathrm{Info}}=\lambda$, where α is the factor in front of the added mutual information term $I_{\phi }\left(X;\tilde{Z}\right)$ and where λ is artificially inserted. It should be emphasised that, for positive $\lambda_{B}$ and $\lambda_{\mathrm{Info}}$, the InfoVAE loss is not generalised by the BIB-AE formalism. The additional $\lambda_{B}\lambda_{\mathrm{Info}}D_{\mathrm{KL}}\left({\tilde{q}}_{\phi }\left(z\right)∥p\left(z\right)\right)$ term is, however, very similar to a term that we will later find in the TURBO formalism, which is due to its mutual information maximisation origin. The success of the InfoVAE framework is a strong incentive for the development of the TURBO formalism.

#### 3.1.2. GAN from BIB-AE Perspectives

On the other side of the widely used yet very related deep learning generative models stand GANs [1]. The principle of a basic GAN is simple as it may be summarised by a decoder network that is trained to map a random input latent sample to an output sample compatible with the data. Typically, the latent space again follows a Gaussian distribution to ensure simple sampling, similar to VAEs. However, since there is no encoder in the usual GAN formulation, there is no need to include any shaping of the latent space by means of some loss terms. The sole role of the decoder is to transform the Gaussian distribution into the data distribution p(x). Therefore, the training objective of a GAN can be expressed as the fourth term of Equation (5)
(8)$$L^{\mathrm{GAN}}\left(\theta \right)=D_{\mathrm{KL}}\left(p\left(x\right)∥{\hat{p}}_{\theta }\left(x\right)\right),$$
where we omit the $\lambda_{B}$ weight as it has no impact here. This term ensures the closeness of the true data distribution p(x) and the generated data distribution ${\hat{p}}_{\theta }\left(x\right)$.

In contrast to the KLD term in the latent space of the VAE formulation, the GAN loss cannot be expressed in a closed form because the data distribution p(x) that the model tries to approximate with ${\hat{p}}_{\theta }\left(x\right)$ is by definition unknown. Facing an intractable KLD is unfortunately a common scenario in machine learning optimisation problems. In practice, the loss is usually replaced by any differentiable metric suited to the comparison of two distributions using samples from both. This includes, but is not limited to, density ratio estimation through a discriminator network and optimal transport through Wasserstein distance approximations. We refer the interested reader to [20,28] for an overview of different methods that allow us to tackle this problem in a practical way.

It is worth noting that, in principle, the reconstructed marginal distribution ${\hat{p}}_{\theta }\left(x\right)$ involves the latent marginal distribution ${\tilde{q}}_{\phi }\left(z\right)$ in its definition. However, since there is no encoder network in the case of a GAN, it must be understood as the true latent distribution ${\tilde{q}}_{\phi }\left(z\right)\equiv p\left(z\right)$, which would correspond to a perfect encoder $q_{\phi }\left(z|x\right)\equiv p\left(z|x\right)$. The TURBO formalism will later provide deeper insights into GANs.

For the sake of completeness, we also quote here the hybrid VAE/GAN loss [29]. It is hybrid in the sense that it uses both the VAE-like latent space regularisation and the GAN-like reconstruction space distribution matching. The VAE/GAN loss can be expressed as the first, third and fourth terms of Equation (5): (9)$$\begin{array}{cc} L^{\mathrm{VAE}/\mathrm{GAN}}\left(\phi ,\theta \right)= & E_{p(x)}\left[D_{\mathrm{KL}}\left(q_{\phi }\left(z|x\right)∥p\left(z\right)\right)\right]\\ \; & -\lambda_{B}\;E_{q_{\phi }\left(x,z\right)}\left[\log p_{\theta }\left(x|z\right)\right]+\lambda_{B}\;D_{\mathrm{KL}}\left(p\left(x\right)∥{\hat{p}}_{\theta }\left(x\right)\right), \end{array}$$
where we do not take into account the refinement of the reconstruction error term proposed in the original work. Indeed, the VAE/GAN loss replaces the likelihood term $p_{\theta }\left(x|z\right)$ with a Gaussian distribution on the outputs of the hidden layers of the discriminator network.

#### 3.1.3. CLUB

Another noticeable extension of BIB-AE is the CLUB model [20]. This model extends the IBN principle by providing a unified generalised framework for information-theoretic privacy models, and by establishing a new interpretation of popular generation, discrimination and compression models. Using our notations, the CLUB objective function can be expressed as
(10)$$L^{\mathrm{CLUB}}\left(\phi ,\theta ,\theta_{s}\right)=I_{\phi }\left(X;\tilde{Z}\right)-\lambda_{B}\;I_{\phi ,\theta }^{x}\left(X;\tilde{Z}\right)+\lambda_{S}\;I_{\phi ,\theta_{s}}^{s}\left(S;\tilde{Z}\right),$$
which is the BIB-AE loss of Equation (4) plus the additional privacy term $\lambda_{S}\;I_{\phi ,\theta_{s}}^{s}\left(S;\tilde{Z}\right)$ controlled by the positive weight $\lambda_{S}$. The S variable denotes the *sensitive attribute* linked to the data X, which has to be kept secret for privacy purposes. The additional minimised term $I_{\phi ,\theta_{s}}^{s}\left(S;\tilde{Z}\right)$ is an upper bound to the mutual information between S and $\tilde{Z}$, taking into account an attacker network with parameters $\theta_{s}$, which aims at inferring the sensitive attribute from the latent variable $\tilde{Z}$. The trade-off between the three terms of the CLUB loss can be understood as an attacker–defender game. Indeed, the first term tries to globally reduce the shared information between the input data and the latent space, while the second and third terms compete in order to keep and get rid of, respectively, the information needed to reconstruct $\hat{X}$ from $\tilde{Z}$ and to infer $\hat{S}$ from $\tilde{Z}$.

### 3.2. Max-Max Game: Or Physically Meaningful Latent Space

So far, we have presented the IBN principle, whose main objective is to find the optimal way to compress data into a latent representation while preserving enough information for the task at hand. What if one does not need any compression of the data? Alternatively, what if the latent space should not be Gaussian in order to facilitate the compression and the sampling? In other words, what if the latent space does not have to be *latent*? Indeed, except if partially removing the information contained in the data serves a particular purpose, one could want to retain all the information and just map the data to a latent space of the desired shape. More precisely, one can get rid of the trade-off expressed in Equation (4) and train a network to only maximise the mutual information between the data and latent spaces. It should be noticed that the oxymoronic phrasing, namely a *physical latent space*, is used to emphasise the questions raised here. Since it is a common name for the intermediate representation in an auto-encoder setting, we choose to keep using the word *latent* for denoting this space.

An AAE [4], for example, is based on an adversarial loss computed in the latent space, very much like the GAN loss in the data space. This is a first hint that this loss term might come from the maximisation of mutual information rather than minimisation. It can also be understood as the second term of Equation (5) but with the opposite sign. This is a second hint that goes towards the same conclusion. In the following section, we will detail the TURBO formalism, which will provide a rigorous explanation of AAEs where the BIB-AE formulation fails to do so.

The formalisms that do not show a bottleneck architecture are also highly relevant in several contexts. Indeed, in many cases, the latent space should not be seen as a mere compression of the data but rather as a physically meaningful representation of them. Figure 3 shows a comparison of two auto-encoders settings, one with a virtual latent space and one with a physical latent space. For example, a virtual latent variable could represent a class label, while a physical latent variable would be a noisy picture captured by some camera. In the physical setting, the complex yet unknown physical measurement is approximated by the parametrised network corresponding to the encoder $q_{\phi }\left(z|x\right)$, which produces the observation $\tilde{z}$ as output. The parametrised decoder $p_{\theta }\left(x|z\right)$ reconstructs back $\hat{x}$ or converts it to a suitable form for further analysis or storage. The latent space variable $\tilde{z}$ is thus defined by the physics of the experiment rather than being fixed to follow a Gaussian, a categorical or any other simple distribution.

One may even compose these settings as shown in Figure 4. For example, the latent variable $\tilde{z}$ can be understood as a generic name encapsulating both a virtual ${\tilde{z}}_{v}$ and a physical ${\tilde{z}}_{p}$ meaning. On the other hand, the encoder and decoder themselves could be composed of their own auto-encoder-like networks, each processing some kind of internal latent variable $y_{e}$ and $y_{d}$, respectively. Combining the BIB-AE and the TURBO formalisms in order to create such deep networks can lead to a very broad family of models.

Treating the latent variable as a second representation of the data in a different space is useful for several applications, as sketched in Figure 5. This is the case, for example, with the optimal-transport-based unfolding and simulation (OTUS) method [30] in the context of high-energy physics. Here, the latent space is the four-momenta of the particles produced in a proton–proton collider experiment, while the data space is the detector response to the passage of these particles. A sketch of the two representations is shown in the top row of Figure 5.

Another example might be the image-to-image translation of a given portion of the sky pictured by two different telescopes. In this case, the latent and data spaces are both images, but one is the image of the sky as seen by the first telescope (e.g., the Hubble Space Telescope), while the other is the image of the sky as seen by the second telescope (e.g., the James Webb Space Telescope). A sketch of the two representations is shown in the middle row of Figure 5.

A third example of a problem where two representations of the data are available is the analysis of copy detection patterns (CDPs) for anti-counterfeiting applications. Templates of CDPs printed on products can be scanned by a device such as a phone camera. The original pattern and the digitally acquired one form a pair of meaningful representations of the same data. A sketch of the two representations is shown in the bottom row of Figure 5.

In summary, in many physical observation or measurement systems, we can consider the measured data as a latent space. The encoder therefore reflects the nature of a measurement or a sampling process. Further extraction of useful information can be considered as decoding the latent measured data and the overall system can be interpreted as a physical auto-encoder. In such a setting, the measured data are in general not Gaussian. Moreover, the data sensors usually being designed to provide as much information as possible about some events or phenomena leads to a maximisation of mutual information between the studied phenomena and their observations rather than a minimisation. There are many situations where the latent space has a physical meaning as relevant as the data space. In such domains, it may be highly relevant to keep as much information as possible, if not all of it, in the latent space by maximising its mutual information with the data space. This is the most important concept leading to the TURBO formalism.

## 4. TURBO

In this section, we present the TURBO framework, which is formulated as a generalised auto-encoder framework. The main ingredient of TURBO is its general loss derived from the maximisation of various lower bounds to several mutual information expressions as opposed to the variational IBN counterpart. Additionally, instead of considering a single-way direction of information flow from x to $\tilde{z}$ and back to $\hat{x}$, TURBO considers a two-way uni-directional information flow that also includes the flow from z to $\tilde{x}$ and back to $\hat{z}$. It is important to note that TURBO interprets the random variables X and Z as following the joint distribution p(x,z) instead of treating them independently as in the BIB-AE formulation. Furthermore, once the general TURBO loss is developed, turning on and off the different terms allows us to recover many existing models, such as AAE [4], GAN [1], WGAN [31], pix2pix [15], SRGAN [32], CycleGAN [16] and even normalising flows [17]. With minor extensions, it also allows us to recover other models such as ALAE [18]. Maybe even more importantly, new models could also be uncovered by using new combinations of the TURBO loss terms. Therefore, the TURBO framework not only summarises existing systems that cannot be explained by the traditional IBN but also creates paths for the development of new ones.

### 4.1. General Objective Function

The starting point of TURBO is to express several forms of the mutual information between the data space and the latent space in the auto-encoder formulation. Three types of mutual information expressions, highlighted in Figure 2, are studied. We consider the mutual information for the real dataset (x,z)∼p(x,z), which is denoted by I(X;Z). We also consider the two evolving mutual information expressions given by the encoded dataset $\left(x,\tilde{z}\right)∼q_{\phi }\left(x,z\right)$ and the decoded dataset $\left(\tilde{x},z\right)∼p_{\theta }\left(x,z\right)$, denoted by $I_{\phi }\left(X;\tilde{Z}\right)$ and $I_{\theta }\left(\tilde{X};Z\right)$, respectively. In the general case, the computation of mutual information in high-dimensional space for real data is infeasible. To make this problem tractable, we introduce four different lower bounds to these expressions. The technical details about the derivations can be found in Appendix C. The objective function of TURBO is based on these lower bounds and consists of their maximisation with respect to the parameters of the encoder and decoder networks.

For convenience, the objective function is translated into a loss minimisation problem. It involves eight terms, whose short notations are defined here for ease of reading: $$\begin{array}{cccc} L_{\tilde{z}}\left(z,\tilde{z}\right) & :=-E_{p(x,z)}\left[\log q_{\phi }\left(z|x\right)\right] & D_{\tilde{z}}\left(z,\tilde{z}\right) & :=D_{\mathrm{KL}}\left(p\left(z\right)∥{\tilde{q}}_{\phi }\left(z\right)\right)\\ L_{\hat{x}}\left(x,\hat{x}\right) & :=-E_{q_{\phi }\left(x,z\right)}\left[\log p_{\theta }\left(x|z\right)\right] & D_{\hat{x}}\left(x,\hat{x}\right) & :=D_{\mathrm{KL}}\left(p\left(x\right)∥{\hat{p}}_{\theta }\left(x\right)\right)\\ L_{\tilde{x}}\left(x,\tilde{x}\right) & :=-E_{p(x,z)}\left[\log p_{\theta }\left(x|z\right)\right] & D_{\tilde{x}}\left(x,\tilde{x}\right) & :=D_{\mathrm{KL}}\left(p\left(x\right)∥{\tilde{p}}_{\theta }\left(x\right)\right)\\ L_{\hat{z}}\left(z,\hat{z}\right) & :=-E_{p_{\theta }\left(x,z\right)}\left[\log q_{\phi }\left(z|x\right)\right] & D_{\hat{z}}\left(z,\hat{z}\right) & :=D_{\mathrm{KL}}\left(p\left(z\right)∥{\hat{q}}_{\phi }\left(z\right)\right). \end{array}$$The four terms in the left column correspond to conditional cross-entropies while the four terms in the right column represent KLDs between the true marginals and the marginals in the latent or reconstruction spaces. It should be noted that every KLD term is forward, meaning that the expected values are taken over the true data distribution. The losses involving such terms are therefore called *mean-seeking* as opposed to *mode-seeking*. While, in general, the choice of the order of the KLD follows empirical observations, the TURBO framework sets it beforehand. However, in practice, the KLD is usually approximated with expressions that often appear to lose this asymmetry property.

The conditional cross-entropy terms reflect the pair-wise relationships while the unpaired KLD terms characterise the correspondences between the distributions. Again, a common choice of the conditional distributions in the cross-entropies is to assume exponential deviations, leading to the $\ell_{2}$-norm or the $\ell_{1}$-norm in the special cases of multi-variate Gaussian or Laplacian, respectively. Therefore, the conditional cross-entropy and KLD terms can be computed in practice, providing the corresponding bound to the considered mutual information terms.

Each auto-encoder has constraints on the latent and reconstruction spaces, but the latent space of TURBO is not *fictitious* as in the IBN framework. It is rather linked to observable variables and that is why, instead of the IBN information minimisation in the latent space, TURBO considers information maximisation.

In contrast to the traditional single-way uni-directional IBN, the TURBO framework considers two flows of information named as *direct* and *reverse* paths. TURBO assumes that the real observable data follow the joint distribution (x,z)∼p(x,z). Therefore, the two-way uni-directional nature of TURBO reflects the fact that this joint distribution can be decomposed in two different ways using the chain rule for probability distributions p(x,z)=p(x)p(z|x)=p(z)p(x|z). Each path of TURBO corresponds to its own auto-encoder setting as shown in Figure 6 and Figure 7. Each auto-encoder consists of two parametrised networks, $q_{\phi }\left(z|x\right)$ and $p_{\theta }\left(x|z\right)$, which are shared between the *direct* and *reverse* paths. Only the order in which they are used is changed.

The *direct* path loss of TURBO corresponding to the encoding of the variable x into $\tilde{z}$ and then the decoding of it back into $\hat{x}$ as shown in Figure 6 is defined as
(11)$$\begin{array}{cc} L^{\mathrm{direct}}\left(\phi ,\theta \right)\; & =-I_{\phi }^{z}\left(X;Z\right)-\lambda_{D}I_{\phi ,\theta }^{x}\left(X;\tilde{Z}\right)\\ \; & =L_{\tilde{z}}\left(z,\tilde{z}\right)+D_{\tilde{z}}\left(z,\tilde{z}\right)+\lambda_{D}L_{\hat{x}}\left(x,\hat{x}\right)+\lambda_{D}D_{\hat{x}}\left(x,\hat{x}\right), \end{array}$$
where $\lambda_{D}$ is a hyperparameter controlling the relative importance of the two mutual information bounds $I_{\phi }^{z}\left(X;Z\right)$ and $I_{\phi ,\theta }^{x}\left(X;\tilde{Z}\right)$ derived in Section C.1 and Section C.2, respectively. The formulation in Equation (11) is expressed in terms of a *loss* that is typically minimised in machine learning applications. That is why both terms have a minus sign in front of them. It should still be non-ambiguously considered as the maximisation of mutual information. The encoder part of the *direct* path loss ensures that the latent space variable $\tilde{z}$ produced by the encoder $q_{\phi }\left(z|x\right)$ matches its observable counterpart z for a given pair (x,z), according to the $L_{\tilde{z}}\left(z,\tilde{z}\right)$ term, while their marginals should be as close as possible according to the $D_{\tilde{z}}\left(z,\tilde{z}\right)$ term. The decoder part of the loss ensures the pair-wise correspondence between the reconstructed variable $\hat{x}$ produced by the decoder $p_{\theta }\left(x|z\right)$ from $\tilde{z}$, according to the term $L_{\hat{x}}\left(x,\hat{x}\right)$, while the $D_{\hat{x}}\left(x,\hat{x}\right)$ term guarantees the matching of the reconstructed and true data distributions.

The *reverse* path loss of TURBO corresponding to the decoding of the variable z into $\tilde{x}$ and then the encoding of it back into $\hat{z}$ as shown in Figure 7 is defined as
(12)$$\begin{array}{cc} L^{\mathrm{reverse}}\left(\phi ,\theta \right)\; & =-I_{\theta }^{x}\left(X;Z\right)-\lambda_{R}I_{\phi ,\theta }^{z}\left(\tilde{X};Z\right)\\ \; & =L_{\tilde{x}}\left(x,\tilde{x}\right)+D_{\tilde{x}}\left(x,\tilde{x}\right)+\lambda_{R}L_{\hat{z}}\left(z,\hat{z}\right)+\lambda_{R}D_{\hat{z}}\left(z,\hat{z}\right), \end{array}$$
where $\lambda_{R}$ is another hyperparameter controlling the relative importance of the two mutual information bounds $I_{\theta }^{x}\left(X;Z\right)$ and $I_{\phi ,\theta }^{z}\left(\tilde{X};Z\right)$ derived in Section C.3 and Section C.4, respectively. The interpretation of these four terms is analogous to the *direct* path.

The complete TURBO loss is finally defined as the weighted sum of the *direct* and *reverse* paths losses
(13)$$L^{\mathrm{TURBO}}\left(\phi ,\theta \right)=L^{\mathrm{direct}}\left(\phi ,\theta \right)+\lambda_{T}L^{\mathrm{reverse}}\left(\phi ,\theta \right),$$
where a hyperparameter $\lambda_{T}$ controls the relative importance of the two terms. It is worth noting that the full loss contains four bounds to mutual information expressions that involve both network sets of parameters ϕ and θ. In general, the optimal parameters that would maximise these four terms separately do not coincide. Therefore, the global solution of the complete optimisation problem usually shows deviations from the said optimal parameters, which is strongly dependent on the trade-off weights that balance the different bounds. Moreover, in practice, it is often impossible to calculate all the terms of the TURBO loss due to the nature of the data considered. For example, pairwise comparisons are particularly affected when there is no labelled correspondence between the X and Z variable domains. This results in multiple relevant architectures, whose objective functions lead to different optimal parameters.

### 4.2. Generalisation of Many Models

The complete loss being defined, we can now relate it to several well-known models. This means that the TURBO framework is a generalisation of these models that gives a uniform interpretation of their respective objective functions and creates a common basis towards explainable machine learning.

#### 4.2.1. AAE

The inability of the BIB-AE framework to explain the celebrated family of AAEs [4] was among the many motivation factors in developing the new TURBO framework. Indeed, the AAE loss can now simply be expressed as the second and third terms of Equation (11):(14)$$L^{\mathrm{AAE}}\left(\phi ,\theta \right)=D_{\tilde{z}}\left(z,\tilde{z}\right)+\lambda_{D}L_{\hat{x}}\left(x,\hat{x}\right),$$
where we recognise the adversarial loss in the latent space Z and the reconstruction loss in the data space X. The latent space distribution is controlled by the imposed prior p(z). Figure 8 shows a schematic representation of the AAE architecture in the TURBO framework. Notice that the AAE, as a representative of the classical auto-encoder family, only uses the *direct* path.

#### 4.2.2. GAN and WGAN

As stated previously, GANs [1] are included into the BIB-AE formalism via a convoluted explanation of the distributions used to compute the adversarial loss. On the other hand, they can be easily expressed in the TURBO framework using only the second term of Equation (12):(15)$$L^{\mathrm{GAN}}\left(\theta \right)=D_{\tilde{x}}\left(x,\tilde{x}\right),$$
which much more naturally involves the data marginal distribution p(x) and the approximated marginal distribution ${\tilde{p}}_{\theta }\left(x\right)$. In this formulation, the Z space is a placeholder used to represent any input to the decoder. It can be pure random noise, as for the StyleGAN model [33], or also include additional information such as class labels, as for the BigGAN [34] and StyleGAN-XL [35] large-scale models, which nowadays still compete with other modern frameworks [36,37]. As stated previously, the KLD term $D_{\tilde{x}}\left(x,\tilde{x}\right)$ can be replaced by Wasserstein distance approximations, leading to the so-called Wasserstein GAN (WGAN) model [31]. Figure 9 shows a schematic representation of the GAN architecture in the TURBO framework. Notice that the classical GAN family does not make use of an encoder network and thus is better interpreted in the *reverse* path.

#### 4.2.3. pix2pix and SRGAN

The pix2pix [15] and SRGAN [32] architectures are conditional GANs initially developed for image-to-image translation and image super-resolution, respectively. During training, both pix2pix and SRGAN assume the presence of *N* training pairs ${\left\{x_{i},z_{i}\right\}}_{i=1}^{N}$, allowing one to use a paired loss for the translation network or the decoder optimisation. However, since the typically used losses, such as $\ell_{2}$-norm, do not cope with the statistical features of natural images, such a decoder produces poor results. This is why the training loss is additionally complemented by an adversarial term, the goal of which is to ensure that the translated images $\tilde{x}$ are on the same manifold as the training data x. Such an adversarial loss does not require paired data and is similar to the GAN family.

The considered paired systems can be expressed in the TURBO framework using the first and the second terms of Equation (12):(16)$$L^{\mathrm{pix}2\mathrm{pix}}\left(\theta \right)=L_{\tilde{x}}\left(x,\tilde{x}\right)+D_{\tilde{x}}\left(x,\tilde{x}\right),$$
where the Z space now represents the image to be translated, while the additional random noise also used as input has to be implicitly understood. Figure 10 shows a schematic representation of the pix2pix and the SRGAN architectures in the TURBO framework. It is important to note that pix2pix and SRGAN do not consider the back reconstruction of $\hat{z}$ from the generated data $\tilde{x}$ as present in the full TURBO framework. Nevertheless, the fact that such systems have been proposed in the prior art and have produced state-of-the-art results for image-to-image translation and image super-resolution problems motivated us to consider them from the point of view of the TURBO generalisation.

#### 4.2.4. CycleGAN

The CycleGAN [16] architecture is an image-to-image translation model as well, but designed for unpaired data. It can be thought of as an AAE trained in two ways, where the X and Z spaces represent the two domains to be translated into each other. The CycleGAN loss can therefore be expressed in the TURBO framework using the second and the third terms of Equation (11) plus the second and the third terms of Equation (12):(17)$$L^{\mathrm{CycleGAN}}\left(\phi ,\theta \right)=D_{\tilde{z}}\left(z,\tilde{z}\right)+\lambda_{D}L_{\hat{x}}\left(x,\hat{x}\right)+\lambda_{T}D_{\tilde{x}}\left(x,\tilde{x}\right)+\lambda_{T}\lambda_{R}L_{\hat{z}}\left(z,\hat{z}\right),$$
where $\lambda_{T}=1$ and $\lambda_{D}=\lambda_{R}$ in the original loss formulation. Figure 11 shows a schematic representation of the CycleGAN architecture in the TURBO framework.

#### 4.2.5. Flows

In order to map a tractable base distribution to a complex data distribution, normalising flows [17] learn a series of invertible transformations, creating an expressive deterministic invertible function. The optimal function is usually found by maximising the likelihood of the data under the flow transformation. Sampling points from the base distribution and applying the transformation yields samples from the data distribution. Flows also allow one to evaluate the likelihood of a data sample by evaluating the likelihood of the corresponding base sample given by the inverse transformation.

At first look, normalising flows do not seem to fit into an auto-encoder framework. However, a flow can be thought of as a special case of an auto-encoder where the decoder is the deterministic parametrised invertible function, denoted by T(z), while the encoder is its inverse $T^{-1}\left(x\right)$. Actually, the very principle of an auto-encoder is precisely to approximate this ideal case, usually denoting $\tilde{z}=f_{\phi }\left(x\right)=T^{-1}\left(x\right)$ for the encoder output and $\hat{x}=g_{\theta }\left(\tilde{z}\right)=T\left(\tilde{z}\right)$ for the decoder output. The approximated conditional distributions defined by Equations (1) and (2) thus read $p_{\theta }\left(x|z\right)=\delta \left(x-T\left(z\right)\right)$ and $q_{\phi }\left(z|x\right)=\delta \left(z-T^{-1}\left(x\right)\right)$, where δ(·) is the Dirac distribution. Moreover, minimising the KLD between the true and approximated data marginal distributions is equivalent to maximising the likelihood of the data under the flow transformation [38]. The flow loss can therefore be expressed in the TURBO framework using only the second term of Equation (12):(18)$$L^{\mathrm{Flow}}\left(\theta \right)=D_{\tilde{x}}\left(x,\tilde{x}\right),$$
which looks very much like the GAN loss of Equation (15). The differences reside in the way the KLD is computed or approximated, and in the parametrisations of the encoder and the decoder. The Z and X variables represent the base and data samples, respectively. Figure 12 shows a schematic representation of the flow architecture in the TURBO framework.

### 4.3. Extension to Additional Models

In addition to the aforementioned models, other architectures can be expressed in the TURBO framework, provided some minor extensions are made. We give here an example of such an extension.

#### ALAE

The adversarial latent auto-encoder (ALAE) [18] is a model that tries to leverage the advantages of GANs, still using an auto-encoder architecture for better representation learning. The main novelty is to exclusively work in the latent space in order to disentangle this data representation as much as possible. The aim is to facilitate the manipulation of the latent space in downstream tasks, keeping a high quality of the generated data. The ALAE loss can be expressed in the TURBO framework using the third and a minor modification of the fourth terms of Equation (12):(19)$$L^{\mathrm{ALAE}}\left(\phi ,\theta \right)=L_{\hat{z}}\left(z,\hat{z}\right)+{\bar{D}}_{\hat{z}}\left(\tilde{z},\hat{z}\right).$$This last term allows for cross-communication between the *direct* and the *reverse* paths. The derivation of this modified term, still based on mutual information maximisation, is detailed in Appendix D. Figure 13 shows a schematic representation of the ALAE architecture in the TURBO framework.

## 5. Applications

In this section, we present several applications of the TURBO framework in studies about different domains. We highlight that these are summary presentations aiming to showcase how the method can be used and to demonstrate its potential. The complete studies as well as all the details are left to dedicated papers. These applications are somewhat disconnected, and TURBO is applied to diverse data with varying dimensionalities and statistics. Nevertheless, as reported in the corresponding studies, TURBO has additional benefits beyond interoperability, such as a superior performance with respect to the models compared, as well as more stable and more efficient training.

### 5.1. TURBO in High-Energy Physics: Turbo-Sim

The TURBO formalism has been successfully applied to a problem of particles into particles transformation in high-energy physics through the Turbo-Sim model [39]. The task is to transform the real four-momenta of a set of particles created by the collision of two protons in a collider experiment into the observed four-momenta of the particles captured by detectors, and vice versa. A clever interpretation of the problem is to think of the real and the observed spaces as two different representations of the same physical system, and to consider them as the Z and X spaces, respectively [30,40]. In such a case, the Z and X spaces are both physically meaningful and maximising the mutual information between them is highly relevant, making the TURBO formalism a natural choice for the problem.

The complete TURBO formalism as depicted in Figure 6 and Figure 7 and formalised in Equation (13) is implemented in the Turbo-Sim model and compared to the OTUS model [30], the former being composed of two fully connected dense networks as the encoder and decoder. Moreover, we do not implement any of the physical constraints considered in the OTUS model. An example of the distributions generated by the TURBO model is shown in Figure 14 and a subset of the metrics used to evaluate the model is shown in Table 2. One can observe that the method is able to give good results up to uncertainties and even outperforms the OTUS method for several crucial observables. It is worth emphasising that the Turbo-Sim model uses very basic internal encoder and decoder networks, showcasing the strength of the TURBO formalism on its own.

### 5.2. TURBO in Astronomy: Hubble-to-Webb

The advanced TURBO framework has been used in the domain of astronomy, specifically for sensor-to-sensor translation. The challenge involves using TURBO as an image-to-image translation framework to generate simulated images of the James Webb Space Telescope from observed images of the Hubble Space Telescope and vice versa. This application of TURBO, concisely called Hubble-to-Webb, is conducted on paired images of the galaxy cluster SMACS 0723.

In Figure 15, we showcase a side-by-side comparison of astronomical imagery. Several additional demos of Hubble images translated into Webb images by various models are available at https://hubble-to-webb.herokuapp.com/ (accessed on 20 September 23). The leftmost image is sourced from the Hubble Space Telescope, providing us with a crisp and detailed depiction of a celestial region. This Hubble image serves as the input to our TURBO image-to-image translation model. The middle image presents the target representation, captured by the James Webb Space Telescope. The main objective of our model is to predict this high-fidelity Webb image using the Hubble input. On the right, we observe the image generated by the TURBO model, which displays a commendable attempt to replicate the intricate features of the actual Webb image. It is evident from the generated image that the TURBO model has made notable strides in bridging the differences between the two telescopes’ observational capacities.

A comparison is made between three methods, namely CycleGAN, pix2pix and TURBO. The same network architecture is used for all three methods; only the objective function is changed accordingly to reflect Equations (13), (16) and (17). The TURBO approach shows remarkable efficacy, demonstrating a state-of-the-art performance in terms of both the learned perceptual image patch similarity (LPIPS) and the Fréchet inception distance (FID) metrics within the domain of sensor-to-sensor translation, as compared to traditional image-to-image translation frameworks. The results of the Hubble-to-Webb translation are detailed in Table 3. Upon examination of the results table, it can be observed that TURBO supersedes other methods in terms of both the LPIPS and the FID metrics, while being competitive for the mean squared error (MSE), the structural similarity (SSIM) and the peak signal-to-noise ratio (PSNR) metrics. These metrics, although not directly related to per-pixel accuracy, are well-established indicators of image fidelity.

### 5.3. TURBO in Anti-Counterfeiting: Digital Twin

The TURBO framework has been employed to model the printing-imaging channel by leveraging a machine-learning-based *digital twin* for CDPs [41]. CDPs serve as a modern anti-counterfeiting technique in numerous applications. The process involves printing a highly detailed digital template z using an industrial high-resolution printer, resulting in a printed template x. The goal of the model is to accurately estimate the complex stochastic process of printing and to generate predictions $\tilde{x}$ of how a digital template would appear once printed, as well as to reverse the process and predict the original digital template $\tilde{z}$ from the printed one.

The same three methods, namely CycleGAN, pix2pix and TURBO, are also compared in this study. Network architectures are shared by the three methods and the objective functions of Equations (13), (16) and (17) constitute again the key differences between them. The study demonstrates that, regardless of various architectural factors, discriminators and hyperparameters, the TURBO framework consistently outperforms widely used image-to-image translation models. A subset of the results is provided in Table 4. The TURBO model shows better results in almost all metrics, staying competitive in the other. In addition, a UMAP projection [42] of real and generated samples is shown in Figure 16. We can see that synthetic samples $\tilde{z}$ and $\tilde{x}$ are close to the corresponding real ones z and x, respectively. We can also observe two distinct clusters, one for digital and one for printed templates.

In Figure 17, we show a visual comparison of template images. The image on the left is a randomly selected digital template, which acts as the input of our TURBO image-to-image translation model. The middle image is the corresponding printed template captured by a scanner. On the right, we display the image generated by the TURBO model. Visually, the synthetic sample looks almost indistinguishable from its real counterpart, meaning that the TURBO model is meritoriously capable of replicating the stochastic printing-capturing process.

In a recent extension of the work, it is shown that the TURBO framework outperforms the other methods not only on data acquired by a scanner but also on data captured using mobile phones. This demonstrates the robustness and versatility of the TURBO approach across different acquisition devices, highlighting its effectiveness in handling different scenarios and its superiority over traditional methods for both high-resolution scanner data and mobile phone data.

## 6. Conclusions

In this work, we have presented a new formalism, called TURBO, for the description of a class of auto-encoders that do not require a bottleneck structure. The foundation of this formalism is the maximisation of the mutual information between different representations of the data. We argue that this is a powerful paradigm that is worth considering in the design of many machine learning models in general. Indeed, we have shown that TURBO can be used to derive a number of existing models and that simple extensions also based on mutual information maximisation can lead to even more models. We have also highlighted several practical use cases where the TURBO formalism is either state-of-the-art or competitive with other models, demonstrating its versatility and robustness.

Our formulation of TURBO is based on the optimisation of multiple lower bounds to several mutual information terms, but it is important to note that other decompositions of such terms exist. We believe that many more modern machine learning architectures can be interpreted as maximising some form of mutual information. For example, SSL methods are not convincingly described by the IBN principle, and their understanding could certainly benefit from the new perspective provided by the TURBO formalism. Moreover, although general enough to allow for any stochastic neural network design, how to meaningfully bring stochasticity into the TURBO framework is left to future work. Another direct extension of the work presented in this paper would be to test TURBO in other relevant applications, namely any problem for which two modalities of the same underlying physical phenomenon are available. We leave the exploration of these ideas to future work and hope that our study will inspire further research in this direction as well, since having a common and interpretable general theory of deep learning is key to its comprehension.

## Figures and Tables

**Figure 1 entropy-25-01471-f001:**
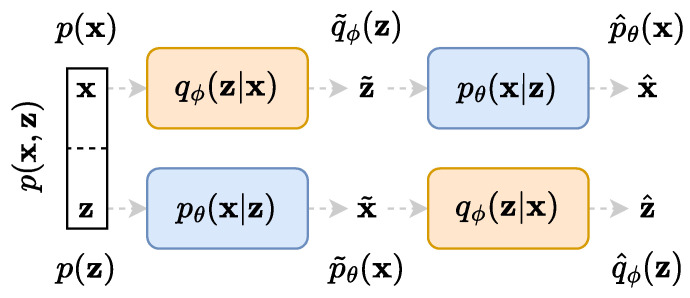
All considered random variable manifolds and the related notations for their probability distributions. The upper part of the diagram is an auto-encoder for the random variable X while the lower part is a symmetrical formulation for the random variable Z. The two random variables X and Z might be independent, so p(x,z)=p(x)p(z).

**Figure 2 entropy-25-01471-f002:**
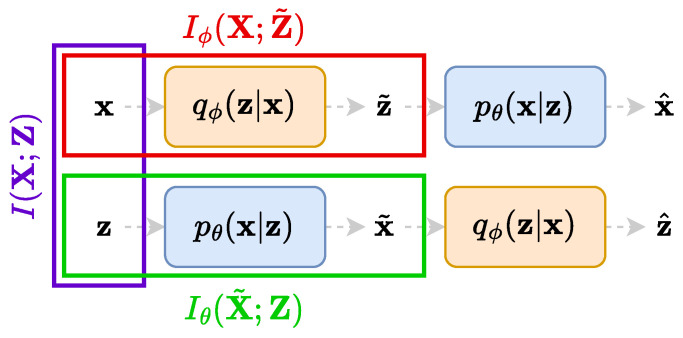
The notations for the mutual information computed between different random variables. The leftmost purple rectangle highlights the true mutual information between X and Z. The upper red and the lower green rectangles highlight the mutual information when the joint distribution is approximated by $q_{\phi }\left(x,z\right)$ and $p_{\theta }\left(x,z\right)$, respectively.

**Figure 3 entropy-25-01471-f003:**
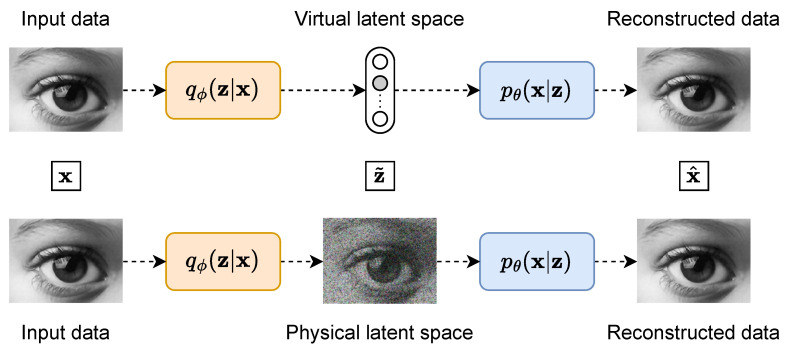
Auto-encoders with a virtual latent space or a physical latent space. In the virtual setting, the latent variable $\tilde{z}$ does not have any physical meaning, while in the physical setting, this latent variable represents a part of the physical observation/measurement chain.

**Figure 4 entropy-25-01471-f004:**
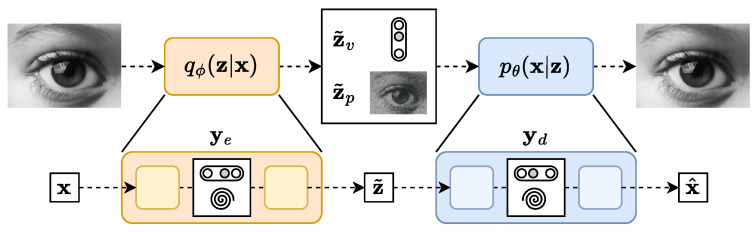
Composition of different latent space settings and several auto-encoder-like networks as internal components of a global auto-encoder architecture. The global latent variable $\tilde{z}$ can contain both virtual ${\tilde{z}}_{v}$ and physical ${\tilde{z}}_{p}$ parts. The global encoder and decoder can be nested auto-encoders with internal latent variables of any kind $y_{e}$ and $y_{d}$, respectively.

**Figure 5 entropy-25-01471-f005:**
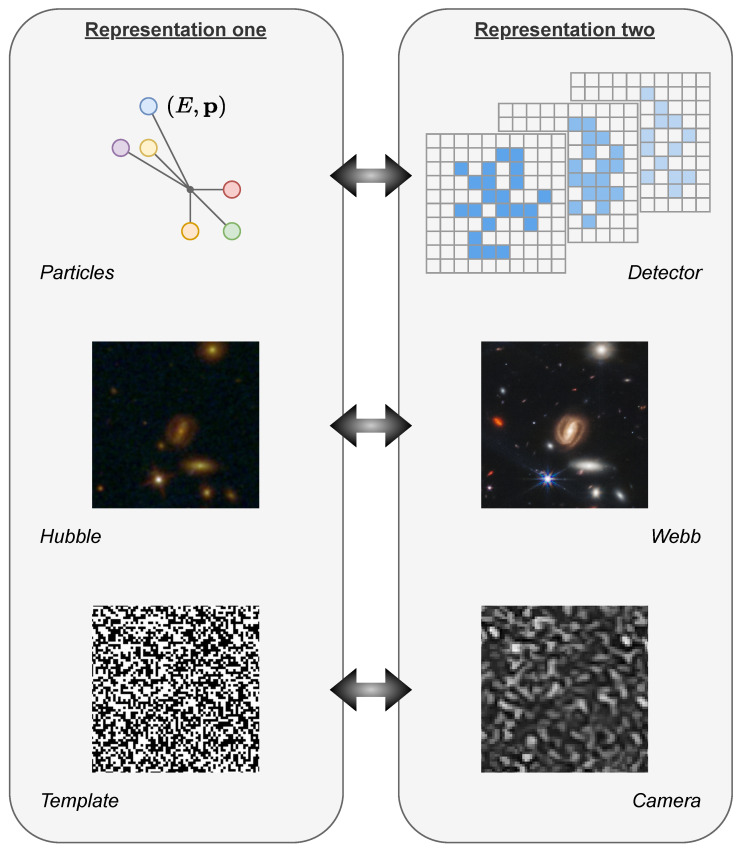
Three different applications that fit into the TURBO formalism. For each domain, two representations of the data are shown, each of which can be associated with one of the two spaces considered, given by the variables X and Z. The top row shows a high-energy physics example, where particles with given four-momenta are created in a collider experiment and detected by a detector. The middle row shows a galaxy imaging example, where two pictures of the same portion of the sky are taken by two different telescopes. The bottom row shows a counterfeiting detection example, where a digital template is acquired by a phone camera.

**Figure 6 entropy-25-01471-f006:**
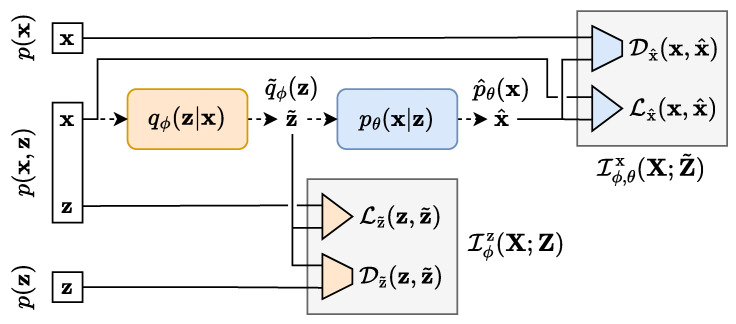
The *direct* path of the TURBO framework. Samples from the X space are encoded following the $q_{\phi }\left(z|x\right)$ parametrised conditional distribution. A reconstruction loss term and a distribution matching loss term can be computed here. Then, the latent samples are decoded following the $p_{\theta }\left(x|z\right)$ parametrised conditional distribution. Another pair of reconstruction and distribution matching loss terms can be computed at this step.

**Figure 7 entropy-25-01471-f007:**
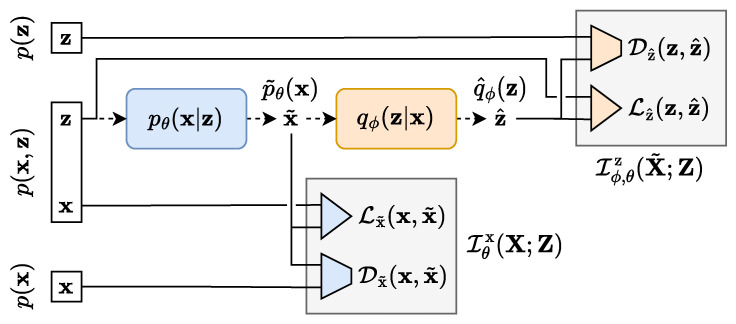
The *reverse* path of the TURBO framework. Samples from the Z space are decoded following the $p_{\theta }\left(x|z\right)$ parametrised conditional distribution. A reconstruction loss term and a distribution matching loss term can be computed here. Then, the latent samples are decoded following the $q_{\phi }\left(z|x\right)$ parametrised conditional distribution. Another pair of reconstruction and distribution matching loss terms can be computed at this step.

**Figure 8 entropy-25-01471-f008:**
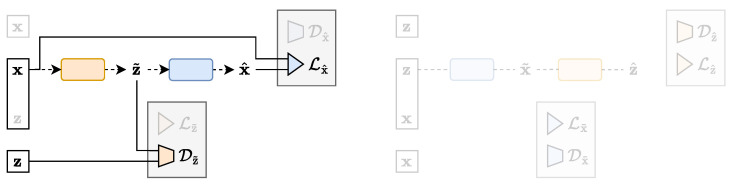
The AAE architecture expressed in the TURBO framework.

**Figure 9 entropy-25-01471-f009:**
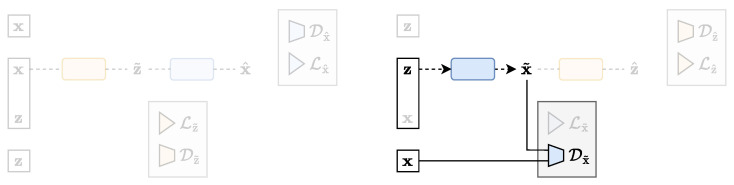
The GAN architecture expressed in the TURBO framework.

**Figure 10 entropy-25-01471-f010:**
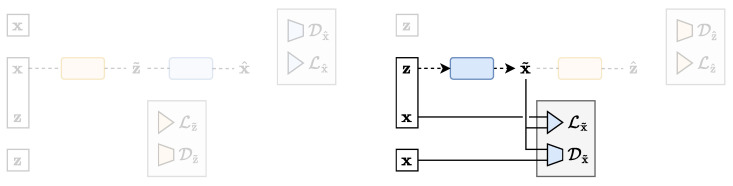
The pix2pix and SRGAN architectures expressed in the TURBO framework.

**Figure 11 entropy-25-01471-f011:**
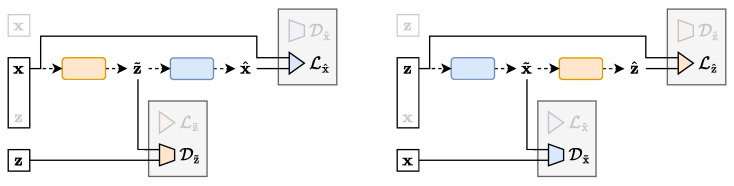
The CycleGAN architecture expressed in the TURBO framework.

**Figure 12 entropy-25-01471-f012:**
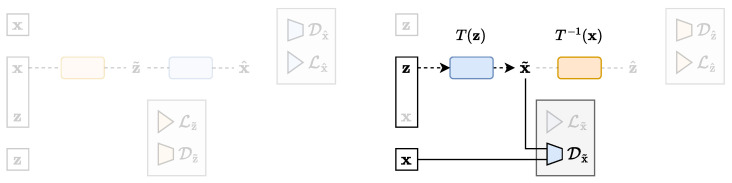
The flow architecture expressed in the TURBO framework.

**Figure 13 entropy-25-01471-f013:**
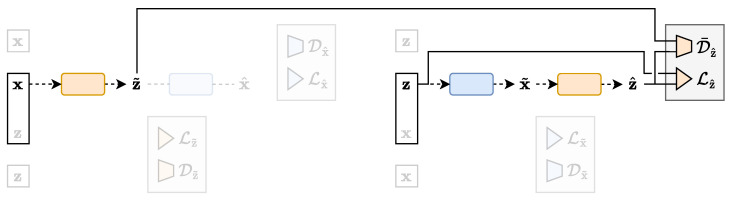
The ALAE architecture expressed in the TURBO framework.

**Figure 14 entropy-25-01471-f014:**
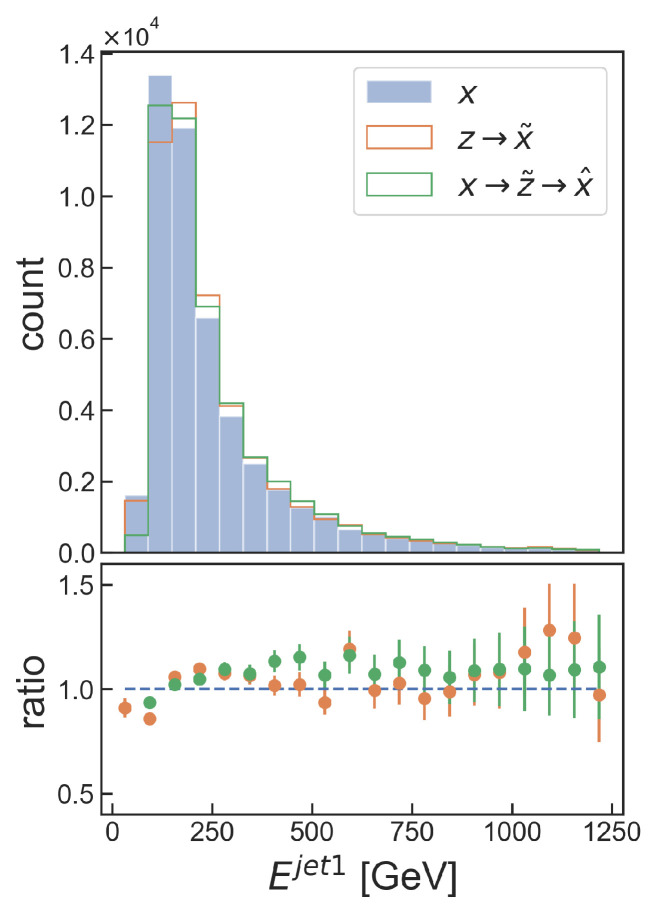
Selected example of distributions generated by the Turbo-Sim model. The histogram shows the distributions of the energy of a given observed particle, which, here, is a shower created by a chain of disintegration called *jet*, for the specific process of top-quark pair production. The blue bars correspond to the original data simulation, the orange line corresponds to the Turbo-Sim transformation from the real particle and the green line corresponds to the Turbo-Sim auto-encoded reconstruction.

**Figure 15 entropy-25-01471-f015:**
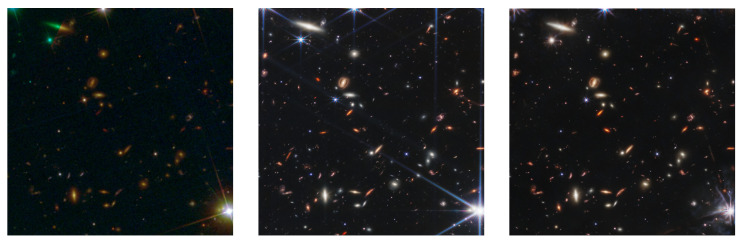
Comparison of images captured by the Hubble Space Telescope (**left**), the James Webb Space Telescope (**middle**) and generated by our TURBO image-to-image translation model (**right**).

**Figure 16 entropy-25-01471-f016:**
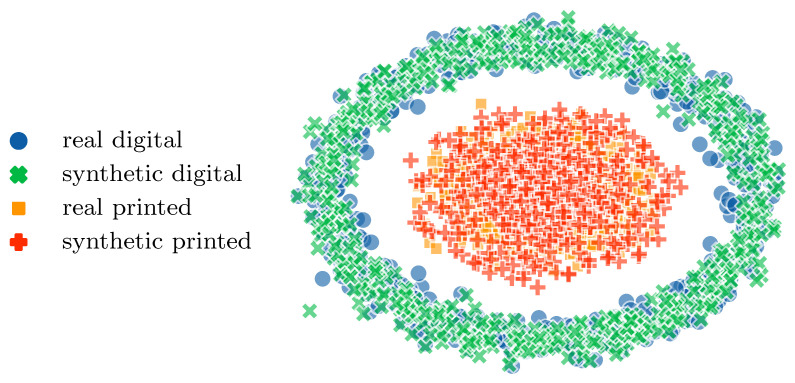
UMAP visualisation of synthetically generated digital and printed templates $\tilde{z}$ and $\tilde{x}$, respectively, superimposed on the corresponding real counterparts z and x.

**Figure 17 entropy-25-01471-f017:**
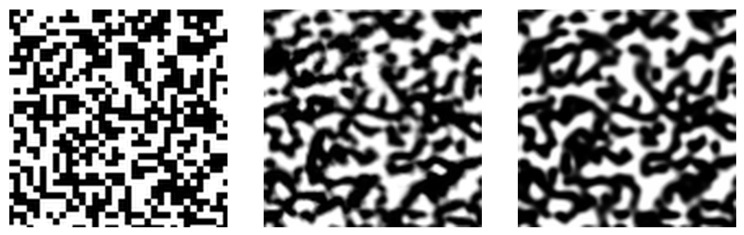
Comparison of a digital template (**left**), printed template (**middle**) and estimation generated by our TURBO image-to-image translation model (**right**). For better visualisation, we display a centrally cropped region that is equal to a quarter of the dimensions of the full image.

**Table 1 entropy-25-01471-t001:** A summary of the main differences between the BIB-AE and the TURBO frameworks.

	BIB-AE	TURBO
Paradigm	**Minimising** the mutual information between the input space and the latent space, while maximising the mutual information between the latent space and the output space	**Maximising** the mutual information between the input space and the latent space, and maximising the mutual information between the latent space and the output space
	**One-way** encoding	**Two-way** encoding
	Data and latent space distributions are considered **independently**	Data and latent space distributions are considered **jointly**
Targeted tasks	Data compression, privacy, classification Representation learning	Linking relevant modalities Transcoding/translation between modalities
Advantages	Theoretical basis for both supervised and unsupervised tasks Allows for easy sampling	Interpretable latent space Seamlessly handles paired, unpaired and partially paired data The encoder can represent a physical system, while the decoder can represent a learnable model
Drawbacks	Not suited for data translation Enforces a distribution for the latent space Struggles to map discontinuous data distributions to continuous latent space distributions	More hyperparameters to tune More modules increases training complexity
Particular cases	VAE, GAN, VAE/GAN	AAE, GAN, pix2pix, SRGAN, CycleGAN, Flows
Related models	InfoVAE, CLUB	ALAE

**Table 2 entropy-25-01471-t002:** Selected subset of the metrics used to evaluate the Turbo-Sim model. The table shows the Kolmogorov–Smirnov distance [$\times 10^{-2}$] between the original data simulation and samples generated by the model. A lower value means a higher accuracy and **bold** highlights the best value per observable. One observable is shown per space. The energy of a real particle, a b-quark, is shown for the Z space, while the energy of the leading jet is shown for the X space. The Rec. column corresponds to unstable particles decaying into the real ones before flying through the detectors to be observed. The observables of these particles must be reconstructed from the combinations of the observed ones. Therefore, the quantity assesses whether the model has learnt the correlations between the variables well enough to make predictions about the underlying physics. In this specific process, two top-quarks are initially produced, and the observable is the invariant mass of the pair.

	*Z* space	*X* space	Rec. space
**Model**	$E^{b}$	$E^{\mathrm{jet}1}$	$m_{\mathrm{tt}}$
Turbo-Sim	3.96	**4.43**	**2.97**
OTUS	**2.76**	5.75	15.8

**Table 3 entropy-25-01471-t003:** Hubble-to-Webb sensor-to-sensor computed metrics. All results are obtained on a validation set of the Galaxy Cluster SMACS 0723. **Bold** highlights the best value per metric.

Model	MSE ↓	SSIM ↑	PSNR ↑	LPIPS ↓	FID ↓
CycleGAN	0.0097	0.83	20.11	0.48	128.1
pix2pix	**0.0021**	**0.93**	**26.78**	0.44	54.58
TURBO	0.0026	0.92	25.88	**0.41**	**43.36**

**Table 4 entropy-25-01471-t004:** Digital twin estimation results. The performances of the models are evaluated on a test split of a dataset acquired by a scanner. The Hamming metric corresponds to the Hamming distance between the z and $\tilde{z}$ samples, while MSE and SSIM are computed between the x and $\tilde{x}$ samples. The FID metric is calculated in both directions. **Bold** highlights the best value per metric and *italic* is reserved to the case with direct comparison without any processing of the data.

Model	FID${}_{x\to \tilde{z}}$ ↓	FID${}_{z\to \tilde{x}}$ ↓	Hamming ↓	MSE ↓	SSIM ↑
*W/O processing*	*304*	*304*	*0.24*	*0.18*	*0.48*
CycleGAN	3.87	**4.45**	0.15	0.05	0.73
pix2pix	3.37	8.57	0.11	0.05	0.76
TURBO	**3.16**	6.60	**0.09**	**0.04**	**0.78**

## Data Availability

No data has been used nor created for this study. The information about the data relative to the applications in Section 5 is provided in the corresponding papers.

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
