# Peer review of "TURBO: The Swiss Knife of Auto-Encoders"

_entropy, 2023, doi:10.3390/e25101471_

Round 1

Reviewer 1 Report

This paper presents the TURBO framework, which attempts to describe different aspects and types of AEs within the bottleneck paradigm.

This work is interesting both in terms of the TURBO framework as well as in terms of how the different types of AEs are being classified and described. The authors provide adequate mathematical as well as graphical intuition in terms of diagrams. Derivations and further information is provided in the form of an appendix, which is appreciated. 

A few suggestions for improvement: 

(1) The TURBO framework should, in my view, be introduced earlier in the paper

(2) It would be helpful if the authors could add a table or otherwise summarise the differences between the different frameworks in terms of complexity, flexibility, etc. This is in effect described in section 3, but a summary would be a welcome addition. 

(3) I don't generally agree with the statement that the latent space can also be physical, as this contradicts the general definition of latent space, and indeed of the word "latent". This case is made in pages 7, 8. I think this needs rephrasing and/or more discussion/justification.

(4) In the applications section, more information would be ver welcome, even if not all applications are presented. Furthermore, it's not immediately obvious how TURBO, a framework a framework for interpretation,  has been used to drive the implementation of these applications, or it has led to "superior performance", and in comparison to what.

(5) I think conclusions should be expanded and include clear directions for future work, either theoretical or in terms of applications.

Reviewer 2 Report

This manuscript presents the TURBO framework and addresses an important gap in the current literature by providing an alternative to the Information Bottleneck (IBN) principle. Highlighting the limitations of the IBN and introducing a new paradigm is commendable.

Overall, the work presents a significant contribution towards the general understanding of neural network models and data representations.

My specific comments and suggestions aimed at enhancing the clarity and impact of this manuscript are detailed below:

1- Although the narrative is generally comprehensible, the flow can be significantly optimized. For better context and continuity, consider discussing the limitations of the IBN principle earlier in the manuscript. This will set the stage for the introduction of TURBO. Additionally, by page 7, there is still no lucid presentation of TURBO. Despite my extensive knowledge in this domain and familiarity with Professor Voloshynovskiy's previous papers on the information bottleneck, the precise formulation and objective of TURBO remained elusive until Section 4.

2- While reviewing this manuscript, I noticed a significant omission. Having thoroughly examined the prior works of the same authors, specifically "Information bottleneck through variational glasses" (https://arxiv.org/pdf/1912.00830.pdf) and "Bottlenecks CLUB: Unifying Information-Theoretic Trade-Offs Among Complexity, Leakage, and Utility" (https://arxiv.org/pdf/2207.04895.pdf), it's clear to me that the CLUB model serves as an extension of the BIB-AE model. Yet, this manuscript neither references that extension nor acknowledges the "CLUB" acronym from their earlier model. This oversight is perplexing.

Additionally, lines 147 to 148 in Section 3 lack accuracy. Contrary to what's stated, the authors in [20] did not merely study the Information Bottleneck (IB) in the context of privacy. Instead, they broadened the scope of IB to encompass privacy-related issues. I also believe that their study was not limited to just "downstream" tasks.

To conclude, my interpretation suggests that the TURBO model can be seen as a specific instance of the CLUB model, particularly when information complexity constraint (encoding rate) R^z is greater than or equal to H(X). This implies a scenario without compression constraints.

3- The phrase "almost totally random" in line 159 seems ambiguous to me. Consider refining it to "possibly random" or another appropriate term for clarity.

4- Ensure consistency in mathematical notation fonts in your figures to match that of the Entropy journal. Also, observe the LaTeX formatting; particularly, avoid unwanted indentations after equations. In your latex code remove your empty line between the equation and the next line, or use "%" to avoid unwanted indentation.

5- There's potential to refine your notations for clarity and succinctness. Please re-evaluate them.

6- The primary contribution seems to be the detailed elucidation in Section 4. While the results consolidate the team's previous findings, a more structured and solid presentation can significantly enhance the manuscript. The visualizations are commendable, but the manuscript structure could be revisited.

7- While the conclusion aptly encapsulates the contributions, consider elaborating on potential areas or questions for future exploration. The current mention of future research (lines 562-565) is notable. However, providing the research community with tangible challenges or directions can spur rapid advancements in this field.

This manuscript promises a valuable addition to the discourse on auto-encoders and deep learning models. Addressing the provided feedback will further refine the work, making it apt for publication.

Needs minor revision

Round 2

Reviewer 1 Report

Thank you for your work and for your responses. I have no further comments.

Reviewer 2 Report

Upon thorough review of the updated draft, as well as considering feedback from other reviewer and the authors' responses, I find that the paper has significantly improved from its initial submission. In light of these improvements, I wholeheartedly recommend accepting the paper for publication.